# Assessment of publication bias and outcome reporting bias in systematic reviews of health services and delivery research: A meta-epidemiological study

**Abimbola A. Ayorinde**[1]*, **Iestyn Williams**[2], **Russell Mannion**[2], **Fujian Song**[3], **Magdalena Skrybant**[4], **Richard J. Lilford**[1], **Yen-Fu Chen**[1]

1 Warwick Centre for Applied Health Research and Delivery, University of Warwick, Coventry, England, United Kingdom, 2 Health Services Management Centre, University of Birmingham, Birmingham, England, United Kingdom, 3 Department of Population Health and Primary Care, University of East Anglia, Norwich, England, United Kingdom, 4 Institute of Applied Health Research, University of Birmingham, Birmingham, England, United Kingdom

* a.ayorinde.1@warwick.ac.uk

**Data Availability Statement:** We have archived the dataset in Warwick Research Archive Portal and

## Abstract

Strategies to identify and mitigate publication bias and outcome reporting bias are frequently adopted in systematic reviews of clinical interventions but it is not clear how often these are applied in systematic reviews relating to quantitative health services and delivery research (HSDR). We examined whether these biases are mentioned and/or otherwise assessed in HSDR systematic reviews, and evaluated associating factors to inform future practice. We randomly selected 200 quantitative HSDR systematic reviews published in the English language from 2007–2017 from the Health Systems Evidence database (www.healthsystemsevidence.org). We extracted data on factors that may influence whether or not authors mention and/or assess publication bias or outcome reporting bias. We found that 43% (n = 85) of the reviews mentioned publication bias and 10% (n = 19) formally assessed it. Outcome reporting bias was mentioned and assessed in 17% (n = 34) of all the systematic reviews. Insufficient number of studies, heterogeneity and lack of pre-registered protocols were the most commonly reported impediments to assessing the biases. In multivariable logistic regression models, both mentioning and formal assessment of publication bias were associated with: inclusion of a meta-analysis; being a review of intervention rather than association studies; higher journal impact factor, and; reporting the use of systematic review guidelines. Assessment of outcome reporting bias was associated with: being an intervention review; authors reporting the use of Grading of Recommendations, Assessment, Development and Evaluations (GRADE), and; inclusion of only controlled trials. Publication bias and outcome reporting bias are infrequently assessed in HSDR systematic reviews. This may reflect the inherent heterogeneity of HSDR evidence and different methodological approaches to synthesising the evidence, lack of awareness of such biases, limits of current tools and lack of pre-registered study protocols for assessing such biases. Strategies to help raise awareness of the biases, and methods to

have been given this link http://wrap.warwick.ac.uk/131604.

**Funding:** This project is funded by the UK National Institute for Health Research (NIHR) Health Services and Delivery Research Programme (project grant number 15/71/06). https://www.nihr.ac.uk/. AA, MS, RJL and YFC are also supported by the NIHR Collaboration for Leadership in Applied Health Research and Care West Midlands (NIHR CLAHRC WM), now recommissioned as NIHR Applied Research Collaboration West Midlands. The views expressed in this publication are those of the authors and not necessarily those of the NIHR or the Department of Health and Social Care. The funders had no role in study design, data collection and analysis, decision to publish, or preparation of the manuscript.

**Competing interests:** The authors have declared that no competing interests exist.

minimise their occurrence and mitigate their impacts on HSDR systematic reviews, are needed.

## Introduction

Health services and delivery research (HSDR) can be defined as "research that is used to produce evidence on the quality, accessibility and organisation of health services including evaluation of how healthcare organisations might improve the delivery of services" [1]. Whilst clinical research into understanding biochemical mechanisms of diseases and their treatments has to some extent dominated health research, the importance of HSDR is increasingly being recognised [2]. For example, a study examining research grants that could impact upon childhood mortality in low-income countries found that 97% of grants were allocated to developing new health technologies, leading to a potential reduction in child death of about 22%, compared to a potential reduction of 63% from research aimed at improving the delivery and utilization of existing technologies [3]. Such finding suggests that while there is a need for research on effective treatments, there is arguably an even greater need for research on the delivery systems that support front line care [4]. With increasing recognition of the importance of HSDR has come increased scrutiny [5]. As with many other fields of research, systematic reviews have proven to be an important tool for summarising and synthesising the rapidly expanding evidence base. The validity of systematic reviews, however, can be undermined by publication bias, which occurs when the publication or non-publication of research findings is determined by the direction or strength of the evidence [6], and by outcome reporting bias whereby only a subset of outcomes, typically those most favourable, are reported [7]. Consequently, the findings that are published (and therefore more likely to be included in systematic reviews) may differ systematically from those that remain unpublished. This results in a biased summary of the evidence which in turn can impair decision making. In HSDR, this could have substantial implications for population health and resource allocation.

To minimise the potential for such biases, mitigating strategies are often included in the process of systematic reviewing. These include: comprehensive literature searching including attempts to locate grey literature or unpublished studies; assessment of outcome reporting bias of included studies; and assessment of potential publication bias using funnel plots, related regression methods and/or other techniques [8]. The level of adoption of such strategies in systematic reviews has been shown to vary by subject area. For example, a study from 2010 which assessed four categories of systematic review from MEDLINE showed that publication bias was assessed in 21% of treatment intervention reviews, 24% of diagnostic test accuracy reviews, 31% of reviews focusing on association between risk factors and health outcomes, and 54% of genetic reviews assessing association between genes and disease [6]. Another study which examined a random sample of 300 systematic reviews of biomedical research indexed in MEDLINE in February 2014 found that 31% had formally assessed publication bias [9]. However, a study examining the reporting characteristics and methodological quality of 99 systematic reviews of health policy research generated by the Cochrane Effective Practice and Organisation of Care Review Group prior to 2014 reported that only 9% of the reviews explicitly assessed publication bias [10]. These findings suggest that the assessment of publication bias is generally low in systematic reviews of clinical research and may be even lower in HSDR and policy research. More detailed information from a broader range of reviews is required to better understand current practice relating to the assessment of publication bias and outcome reporting bias in HSDR systematic reviews. Against this background, the objectives of this study are to examine whether publication bias and outcome reporting bias are mentioned and/or

assessed in a representative sample of HSDR systematic reviews, and to summarise the methods adopted as well as findings reported or reasons stated for not formally assessing the biases.

We focus on systematic reviews of quantitative HSDR studies that involve evaluation of strength and direction of effects, which can be subject to hypothesis testing. Within this broad category, we sampled two review types:

- Intervention reviews, which aim to evaluate the effectiveness of service delivery interventions. These reviews often include randomised controlled trials (RCTs), other quasi-experimental studies and sometimes uncontrolled before-and-after studies, and

- Association reviews, which evaluate associations between different variables (such as nurse-patient ratio, frequency of patient monitoring and in-hospital mortality) along the service delivery causal chain [4]. Association reviews tend to include mostly observational studies.

While intervention reviews usually set out to examine pre-specified causal relationships between an intervention and designated outcomes, association reviews tend to be exploratory. Consequently, the characteristics (such as inclusion of meta-analysis, number and design of included studies, and the use of systematic review guidelines) of these two types of reviews may differ. We hypothesised that association studies may be more susceptible to publication and outcome reporting biases than intervention studies due to the exploratory nature of most association studies. We therefore investigate whether the practice of assessing these biases and the findings of these assessments differ between HSDR systematic reviews focusing on these two types of studies. In addition, we examine whether awareness and/or assessment of publication and outcome reporting biases is associated with factors other than the nature of the review, such as author's use and journal's endorsement of methodological guidelines for the conduct and reporting of systematic reviews, and journal impact factor [11].

## Methods

We carried out a meta-epidemiological study [12] to estimate the frequency with which publication and outcome reporting bias were considered in systematic reviews and to explore factors associated with consideration of these forms of potential bias. The review was pre-registered in the PROSPERO International prospective register of systematic reviews (2016: CRD42016052366 www.crd.york.ac.uk/prospero/display_record.php?ID=CRD42016052366).

### Sampling strategy

Our initial plan for identifying a sample of HSDR systematic reviews specified in the PROSPERO registration record was to conduct a literature search by using a combination of different information sources and searching methods [13]. Retrieved records would subsequently be screened for eligibility and classified as intervention or association reviews before a random sample was selected. However, the proposed sampling strategy was subsequently deemed not feasible given the large number of systematic reviews that would have to be checked for eligibility before sampling. This is due to the methodological diversity of HSDR-related research and the absence of universally accepted terms through which to search for HSDR systematic reviews. We therefore adopted the alternative method of selecting systematic reviews from the Health Systems Evidence (HSE) database (www.healthsystemsevidence.org) [14]. The HSE is a continuously updated repository of syntheses of research evidence about the governance, financial and delivery arrangements within health systems, and the implementation strategies that can support change in health systems [14]. It covers several databases including Medline and Cochrane Database of Systematic Reviews. With the help of the owner of the database, we downloaded all the available citations of systematic reviews indexed in the HSE as of August

2017 into a Microsoft Excel spreadsheet. The HSE classifies each of the systematic reviews into two groups based on the type of question the reviews address; 'effectiveness' for the systematic reviews concerned with effects and 'other questions'. The reviews classed as effectiveness (n = 4416) served as the sampling frame for the intervention reviews while those classed as 'others' (n = 1505) were used for the association reviews. In order to facilitate random selection of reviews, we assigned a random number to each record using the RAND() function in Excel, sorted the random numbers in ascending order before screening the records for eligibility using pre-specified criteria, as described below, until the desired number of reviews was identified.

## Sample size

We aimed to include 200 systematic reviews in total; 100 reviews of intervention studies and 100 reviews of association studies. This sample size has a statistical power of 80% to detect a 20% difference in the characteristics and findings between the two types of review, assuming a baseline rate of 32%, based on the proportion of Cochrane EPOC reviews in which publication bias was formally assessed or for which partial information was given [10].

## Eligibility criteria

In this study, a systematic review was defined as any literature review which presents explicit statements with regard to research question(s), search strategy and criteria for study selection. We also defined HSDR as research that produces evidence on the quality, accessibility and organisation of health services based on the definition adopted by the United Kingdom's National Institute for Health Research (NIHR) Health Services & Delivery Research Programme [1]. Systematic reviews examining quantitative data and relating to any aspects of HSDR were selected irrespective of whether they included a meta-analysis. To be eligible, the systematic review had to report at least one quantitative effect estimate or a statistical test which could be derived from the studies included in the review. Since contemporary literature are of more relevance to current practice, we included reviews from the last ten years (between2007 to 2017). We excluded records which were: not systematic reviews; not related to HSDR; not concerned with interventions or associations; not examining quantitative data; not published in English language; or were published before the year 2007. We also excluded systematic reviews that are usually classified as health technology assessment (such as those investigating effectiveness and cost effectiveness of clinical interventions) and those classified as clinical or genetic epidemiology (that is, those examining association between risk factors and disease conditions). Where more than one review within the initially selected samples cover overlapping interventions or associations, we included the latest review. This helped to maintain the independence of observations and also capture the contemporary practice.

Sample selection was conducted by one author (AAA) and checked by a second author (YFC). Discrepancies were resolved by discussion and members of the research project management team (in the first instance) and steering committee were consulted when the two authors could not reach an agreement or when generic issues concerning study eligibility criteria were identified.

## Data extraction and classification of review characteristics

Data extraction focused on general systematic review characteristics and components that may influence whether or not authors refer to and/or assess publication bias or outcome reporting bias. Thus, data extracted from each eligible review included:

• key study question(s)

- databases searched

- whether an attempt was made to search grey literature and unpublished reports or whether reasons for not doing this were provided

- design of included studies (whether or not these are confined to controlled trials)

- number of included studies (categorised into <10 and ≥10 based on the minimum number of studies recommended for statistical approaches to assessment of publication bias[15])

- Whether meta-analyses were performed

- whether the use of systematic review related guidelines was reported (we assumed that all Cochrane reviews adhered to the Methodological Expectations of Cochrane Intervention Reviews (MECIR) standards [16] even if not reported by authors)

- whether the use of Grading of Recommendations, Assessment, Development and Evaluations (GRADE) was reported

- any mentioning of publication bias and/or outcome reporting bias

- methods (if used at all) for assessing potential publication bias and/or outcome reporting bias

- findings of assessment of publication bias and/or outcome reporting bias or reasons for not formally assessing these

We planned to categorise the types of journals in which the systematic reviews were published based on subject categories of the Journal Citation Reports (ISI Web of Knowledge, Thomson Reuters) as medical journals, health services research and health policy journals, management and social science journals or others(including grey literature), but discovered substantial overlap in the features between journal types which hindered reliable classification of some journals and in turn would cause difficulty in the interpretation of observations made based on the classification. We discussed this issue with the study steering committee members, who suggested us to use journal endorsement of systematic review guidelines and journal impact factors to characterise the journals instead.

Some journals/media require submitted systematic reviews to follow specific systematic review guidelines, for example, PRISMA statement [17], MOOSE checklist [18], MECIR standards (for Cochrane reviews) [16]. Such guidelines includes items on publication bias and may prompt reviewers to consider publication bias, particularly at the manuscript preparation stage. Based on the information available on journal websites, we categorised the journal/media in which the systematic reviews were published into those which formally endorse specific systematic review guidelines and those that do not (as of year 2018). Targeting prestigious journals for publication may also prompt reviewers to be more rigorous and so we identified the five year impact factors (as of year 2016) for the journal each review was published in from ISI Web of Knowledge, Thomson Reuters. When impact factor was not available on the Web of Knowledge website, it was obtained from other sources such as directly from the journal website. We imputed an impact factor of zero for journals with no impact factors and grey literature (such as theses). One author carried out all the data extraction and the data was independently checked by another author. Any discrepancies were resolved by discussion.

## Quality assessment of included systematic reviews

Each systematic review included in the HSE was assessed independently by two reviewers using the Assessing the Methodological Quality of Systematic Reviews (AMSTAR) and the score was provided within the record for each review [19]. However, five of the selected

systematic reviews had missing AMSTAR scores so two authors independently carried out the quality assessment for them using the same version of the AMSTAR tool as the remaining 195 systematic reviews. Discrepancies were resolved by discussion. Percentage AMSTAR scores were computed for each review taking into account the number of items (the denominator) that was applicable to individual reviews.

## Statistical analysis

Descriptive statistics were used to summarise the characteristics of the selected HSDR systematic reviews, the practice of assessing publication bias and outcome reporting bias among the reviews, and their findings. Differences between association reviews and intervention reviews were explored. We presented confidence intervals to indicate the levels of uncertainty but avoided quoting p values and inferring to statistical significance given the descriptive nature of the study and the large number of exploratory comparisons made.

Three measures related to the awareness and actual practice of assessing publication and outcome reporting biases were evaluated:

1. "mentioned publication bias", that is, authors included statements related to publication bias in their report regardless of whether or not this was accompanied by formal assessment (with explicitly stated methods, e.g. use of funnel plots or comparison with findings from search of study registries known to capture all related studies that have been conducted; the latter is unlikely to be feasible in HSDR);

2. "assessed publication bias", which includes only those reviews where publication bias was formally assessed, and

3. "assessed outcome reporting bias" where authors have assessed outcome reporting bias.

Univariable and multivariable logistic regressions were used to explore review and journal characteristics associated with mentioning/assessment of publication bias and outcome reporting bias in the reviews. The strength of association between these variables and practice of bias assessment was presented as an odds ratio (OR) with 95% confidence intervals.

## Results

### Sampling of HSDR systematic reviews from HSE

We screened 220 of the 4416 systematic reviews classified as 'systematic reviews of effects' in the HSE to obtain 100 eligible systematic reviews of intervention for this study. Reviews were excluded mainly because their topics fell outside our definition of HSDR, such as those considered as public health research and health technology assessments. We screen all 1505 systematic reviews classified as 'systematic reviews addressing other questions' to identify 100 eligible systematic reviews of association for this study. Reviews were excluded because the topics under review fell outside our definitions of HSDR and association studies, and/or because their designs did not include a quantitative component, such as reviews adopting narrative and qualitative synthesis approaches and scoping reviews.

### Characteristics of included intervention and association reviews

The characteristics of the included systematic reviews (100 intervention reviews and 100 association reviews) are shown in Table 1. The majority of the 200 systematic reviews (79%) included at least ten studies but less than a quarter (22%) included a meta-analysis. Ninety of the reviews that did not include meta-analysis provided reasons for this–mainly small number

**Table 1. Characteristics of included reviews and comparison between association and intervention reviews.**

| Characteristics | All [n (%)]<br>n = 200 | Association [%]<br>n = 100 | Intervention [%]<br>n = 100 | Difference [a] (%)<br>(95% CI) |
|---|---|---|---|---|
| Number of included studies (≥10) | 157 (79%) | 86 | 71 | 15 (4 to 26) |
| Meta-analysis included | 43 (22%) | 10 | 33 | -23 (-34 to -12) |
| Included only RCT and controlled trials | 36 (18%) | 1[b] | 35 | -34 (-44 to -24) |
| Searched grey/unpublished literature | 103 (52%) | 52 | 51 | 1 (-13 to 15) |
| Quality assessment performed | 157 (79%) | 70 | 87 | -17 (-28 to -6) |
| Authors reported using GRADE | 23 (12%) | 6 | 17 | -11 (-20 to -2) |
| Authors reported using systematic review reporting guideline | 73 (37%) | 28 | 45 | -17 (-30 to -4) |
| Percentage of positive AMSTAR rating [median (IQR)] | 60% (44%, 73%) | 50% (40%, 65%) | 65% (50%, 82%) | -14 (-20 to -10)[c] |
| Journal impact factor in year 2016 [median (IQR)] | 3.00 (2.26, 5.10) | 2.66 (2.07, 3.39) | 3.55 (2.30, 7.08) | -0.98 (-1.73, -.35)[c] |
| Journal endorses systematic review guideline (as of year 2018) | 140 (70%) | 69 | 71 | -2 (-15 to 11) |
| Publication bias mentioned or assessed | 85 (43%) | 31 | 54 | -23 (-36, -10) |
| Publication bias assessed | 19 (10%) | 5 | 14 | -9 (-17, -1) |
| Outcome reporting bias mentioned and assessed | 34 (17%) | 4 | 30 | -26 (-16, -36) |
| Mentioned or assessed publication bias and/or outcome reporting bias | 95 (48%) | 32 | 63 | -31 (-44, -18) |
| Assessed publication bias and/or outcome reporting bias | 49 (24.5%) | 9 | 40 | -31 (-42, -20) |

[a] Comparison between association and intervention reviews

[b] The systematic review was a meta-regression analysis of randomised controlled trials that focused on identifying factors associated with effective computerised clinical decision support systems.

[c] Hodges-Lehmann difference between medians with 95% CI

of comparable studies and high heterogeneity between studies. Searches of grey/unpublished literature were conducted in 52% of the systematic reviews. Quality assessment of individual studies was performed in 79% of the systematic reviews but only 12% reported using GRADE for assessing the overall quality of evidence. The systematic reviews were of moderate quality with median AMSTAR score of 60% (IQR 44% to 73%). Many of the systematic reviews (70%) were published in journals which endorse PRISMA, although the use of such guidelines were only reported in 37% of them.

We observed notable differences between intervention and association reviews in many of the characteristics assessed. For example, intervention reviews were more likely to: include meta-analysis, inclusion of only controlled trials, carry out quality assessment of included studies, report the use of systematic review reporting guidelines and GRADE, have higher AMSTAR ratings and be published in journals with higher impact factors (Table 1). Conversely, association reviews were more likely to include ten or more studies compared with intervention reviews (86% vs 71%). Only the search of grey literature and being published in journal which endorse systematic review guideline were similar in both intervention and association reviews.

## Publication bias

Eighty-five (43%) of the systematic reviews mentioned publication bias and these included a higher proportion of intervention reviews than association reviews (54% vs 31%). Only about 10% (n = 19/200) formally assessed publication bias through statistical analysis, mostly using funnel plots and related methods. Again, intervention reviews assessed publication bias more frequently compared to association reviews (14% vs 5%; Table 1). Some evidence of publication bias (strictly speaking, evidence of small study effects in most instances) was reported in

**Table 2. Factors associated with mentioning publication bias.**

| Factor | All (n = 200) [n (%)] | Mentioned publication bias | | Univariable OR (95% CI) | Multivariable OR (95% CI) |
| | | Yes (n = 85) [n (%)] | No (n = 115) [n (%)] | | |
|---|---|---|---|---|---|
| Being an intervention (versus association) review | 100 (50%) | 54 (64%) | 46 (40%) | 2.61 (1.47–4.66) | 1.63 (0.85–3.15) |
| Number of included studies | 157 (79%) | 66 (78%) | 91 (79%) | 0.92 (0.46–1.81) | 1.16 (0.53–2.53) |
| Meta-analysis included | 43 (22%) | 32 (38%) | 11 (10%) | 5.71 (2.67–12.21) | 4.02 (1.76–9.15) |
| Included only RCT & controlled trials [a] | 36 (18%) | 20 (24%) | 16 (14%) | 1.90 (0.92–3.94) | |
| Searched grey/unpublished literature | 103 (52%) | 46 (54%) | 57 (50%) | 1.20 (0.68–2.10) | 1.16 (0.60–2.23) |
| Quality assessment performed | 157 (79%) | 75 (88%) | 82 (71%) | 3.02 (1.39–6.54) | 2.08 (0.88–4.90) |
| Authors reported using GRADE | 23 (12%) | 15 (18%) | 8 (7%) | 2.87 (1.15–7.12) | 1.58 (0.57–4.44) |
| Authors reported using a systematic review guideline | 73 (37%) | 40 (47%) | 33 (29%) | 2.21 (1.23–3.97) | 1.35 (0.68–2.70) |
| Journal impact factor in the year 2016 [median (IQR)] | 3.00 (2.26, 5.10) | 3.26 (2.27–6.01) | 2.74 (2.18–4.29) | 1.11 (1.02–1.22) | 1.04 (0.96–1.15) |
| Journal endorses a systematic review guideline (as of the year 2018) | 140 (70%) | 61 (72%) | 79 (69%) | 1.16 (0.67–2.14) | 0.94 (0.46–1.93) |

[a] Not included in multivariable analysis as this factor is strongly correlated with review type (intervention vs association)

five (26%) of the reviews which assessed publication bias. The remaining reviews mostly reported low/no risk of publication bias. One review, which included four studies, constructed a funnel plot but reported that it was not very informative due to small numbers [20]. In five of the systematic reviews, authors reported planning statistical assessment of publication bias but did not carry out the assessment due to the conditions of using funnel plots not being met, especially insufficient number of studies and/or heterogeneity between included studies. [21–25]

**Factors associated with mentioning (including assessing) publication bias.** In the univariable analysis, publication bias was more likely to be mentioned in intervention reviews when compared to association reviews (OR 2.61, 95% CI 1.47–4.66). Reviews which included meta-analysis were more than five times more likely to mention publication bias compared to those with no meta-analysis (OR 5.71, 95% CI 2.67–12.21). Mentioning publication bias appeared to be associated with quality assessment of individual studies, authors reporting the use of GRADE, journal impact factor, and authors reporting the use of systematic review guideline (Table 2). Most of the apparent associations attenuated in the multivariable analysis, indicating some levels of interaction between these factors. Inclusion of meta-analysis remained strongly associated with mentioning publication bias (Table 2).

**Factors associated with assessing publication bias.** Intervention reviews were again more likely to include an assessment of publication bias than association reviews (OR 3.09, 95% CI 1.07–8.95). Of all factors assessed, inclusion of meta-analysis was the factor most strongly associated with assessment of publication bias (OR 112.32, 95% CI 14.35–879.03) in the univariable analysis. Only one of the 19 systematic reviews which assessed publication bias did not carry out a meta-analysis. Assessment of publication bias also appeared to be associated with the inclusion of only RCTs and controlled trials, journal impact factor and authors reporting the use of systematic review guidelines (Table 3). Other factors including number of included studies, search of grey/unpublished literature, quality assessment of individual studies and journal endorsement of systematic review guidelines were not significantly associated with assessment of publication bias. In the multivariable analysis, the pattern of apparent associations largely remained the same, although the relationship between assessment of publication bias and two of the factors (types of review and journal impact factors) diminished after adjusting for other factors (Table 3).

**Table 3. Factors associated with the assessment of publication bias.**

| Factor | All (n = 200) [n (%)] | Assessed Publication bias | | Univariable OR (95% CI) | Multivariable |
| | | Yes (n = 19) [n (%)] | No (n = 181) [n (%)] | | |
| --- | --- | --- | --- | --- | --- |
| Being an intervention review (versus association review) | 100 (50%) | 14 (74%) | 86 (48%) | 3.09 (1.07–8.95) | 0.94 (0.20–4.55) |
| Number of included studies (≥10) | 157 (79%) | 17 (90%) | 140 (77%) | 2.49 (0.55–11.22) | 2.21 (0.32–15.27) |
| Meta-analysis included | 43 (22%) | 18 (95%) | 25 (14%) | 112.32 (14.35–879.03) | 84.65 (9.56–749.49) |
| Included only RCT and controlled trials [a] | 36 (18%) | 7 (37%) | 29 (16%) | 3.06 (1.11–8.42) | |
| Searched grey/unpublished literature | 103 (52%) | 6 (32%) | 97 (54%) | 0.40 (0.15–1.10) | 0.34 (0.08–1.46) |
| Quality assessment performed | 157 (79%) | 18 (95%) | 139 (77%) | 5.44 (0.71–41.96) | 5.29 (0.38–82.82) |
| Authors reported using GRADE | 23 (12%) | 2 (11%) | 21 (12%) | 0.90 (0.19–4.16) | 0.47 (0.07–3.38) |
| Authors reported using systematic review guideline | 73 (37%) | 14 (74%) | 59 (33%) | 5.79 (1.99–16.84) | 5.38 (1.19–24.23) |
| Journal impact factor in the year 2016 [median (IQR)] | 3.00 (2.26,5.10) | 3.85 (2.73,5.76) | 2.94 (2.14,4.98) | 1.09 (1.004–1.18) | 1.01 (0.90–1.13) |
| Journal endorses systematic review guideline (as of the year 2018) | 140 (70%) | 10 (53%) | 130 (72%) | 0.44 (0.17–1.34) | 0.22 (0.04–1.09) |

[a] Not included in multivariable analysis as this factor is strongly correlated with review type (intervention vs association)

## Outcome reporting bias

Thirty-four (17%) of all the systematic reviews mentioned and assessed outcome reporting bias as part of quality assessment of included studies. None of the systematic reviews mentioned outcome reporting bias without assessing it. Again this was more frequent in intervention reviews than in association reviews (30% vs 4%). The majority of the reviews which assessed outcome reporting bias used the Cochrane risk of bias tool (n = 28/34) [26]. Two reviews used the Agency for Healthcare Research and Quality's (AHRQ's) Methods Guide for Effectiveness and Comparative Effectiveness Reviews [27], one used the Amsterdam-Maastricht Consensus List for Quality Assessment, while the remaining three reviews used unspecified or bespoke tools. Of the 34 reviews which assessed outcome reporting bias, 31 reported the findings, while the remaining three did not report the findings despite having reported assessing the bias in the methods section. Of the 31 reviews which reported the findings, 35% (n = 11/31) identified at least one study with high risk of selective outcome reporting, 32% (n = 10/31) judged all included studies to be low risk while the remaining 10 reviews (32%) had at least one study where the authors were unable to judge the risk of bias and were classed as 'unclear'. In three reviews, lack of pre-registered protocols was reported as the reason for judging articles as 'unclear'.[20, 22, 28] In a review in which the review authors explicitly stated that they did not search for study protocols, 13 out of the 19 studies included in the review was judged as 'unclear' with regard to selective outcome reporting.[29]

**Factors associated with assessing outcome reporting bias.** Intervention reviews were about ten times as likely to include an assessment of outcome reporting bias compared to association reviews (OR 10.29, 95% CI 3.47–30.53). Assessment of outcome reporting bias was also strongly associated with authors reporting the use of GRADE (OR 9.66, 95% CI 3.77–24.77) and inclusion of RCTs or controlled trials only (OR 7.74, 95% CI 3.39–17.75). Number of included studies, inclusion of meta-analysis, journal impact factor, journal endorsement of systematic review reporting guidelines and authors reporting the use of systematic review guidelines also appeared to be associated with the assessment of outcome reporting bias (Table 4). The variable relating to quality assessment of individual studies was not included in the regression analysis because all studies which assessed outcome reporting bias performed quality assessment of individual studies. Two variables remained strongly associated with

**Table 4. Factors associated with the assessment of outcome reporting bias.**

| Factor | All (n = 200) [n (%)] | Assessed outcome reporting bias | | Univariable OR (95% CI) | Multivariable |
| | | Yes (n = 34) [n (%)] | No (n = 166) [n (%)] | | |
|---|---|---|---|---|---|
| Being an intervention review (versus association review) | 100 (50%) | 30 (88%) | 70 (42%) | 10.29 (3.47–30.53) | 6.44 (2.01–20.60) |
| Number of included studies | 157 (79%) | 20 (59%) | 137 (83%) | 0.30 (0.14–0.67) | 0.53 (0.20–1.43) |
| Meta-analysis included | 43 (22%) | 13 (38%) | 30 (18%) | 2.81 (1.27–6.23) | 1.73 (0.65–4.59) |
| **Included mainly RCT and controlled trials** [a] | 36 (18%) | 17 (50%) | 19 (12%) | 7.74 (3.39–17.75) | |
| Searched grey/unpublished literature | 103 (52%) | 22 (65%) | 81 (49%) | 1.92 (0.89–4.14) | 1.33 (0.51–3.46) |
| Quality assessment performed [b] | 157 (79%) | 34 (100%) | 123 (74%) | | |
| Authors reported using GRADE | 23 (12%) | 13 (38%) | 10 (6%) | 9.66 (3.77–24.77) | 5.18 (1.61–16.67) |
| Authors reported using systematic review guideline | 73 (37%) | 22 (65%) | 51 (31%) | 4.13 (1.90–8.99) | 1.97 (0.78–4.99) |
| Journal impact factor in the year 2016 [median (IQR)] | 3.00 (2.26, 5.10) | 6.58 (2.63,7.08) | 2.77 (2.11,4.28) | 1.10 (1.01–1.19) | 1.04 (0.95–1.13) |
| Journal endorses systematic review guideline (as of the year 2018) | 140 (70%) | 29 (85%) | 111 (67%) | 2.87 (1.05–7.83) | 1.99 (0.65–6.12) |

[a] Not included in multivariable analysis as this factor is strongly correlated with review type (intervention vs association)

[b] Not included in regression analyses because all reviews which assessed outcome reporting bias performed quality assessment

assessing outcome reporting bias in the multivariable analysis: author reporting the use of GRADE and being an intervention review (Table 4).

## Discussion

We obtained a random sample of 200 quantitative systematic reviews in HSDR and examined their characteristics in relation to assessment of publication bias and outcome reporting bias. Only 10% of the systematic reviews formally assessed publication bias even though 43% mentioned publication bias. The majority of the systematic reviews (83%) neither mentioned nor assessed outcome reporting bias. A higher proportion of the intervention reviews mentioned and assessed both biases compared to the association reviews.

### Strengths and limitations

One of the strengths of the current study is that a broad range of quantitative HSDR systematic reviews was examined. The HSE database, from which the systematic reviews were selected, covers multiple sources of literature, and our selection was neither limited to a single source of literature nor restricted to highly ranked journals as was the case in previous studies.[30–32] Also, study selection and data extraction were carried out by one person and checked by another in order to ensure accuracy and completeness.

We targeted intervention and association reviews with a quantitative component in HSDR as defined earlier in this paper. The concept of intervention reviews matched well with the category of 'systematic reviews of effects' in the HSE database where we drew our sample. However, clearly delineating association reviews and identifying those incorporating some quantitative components have proven challenging. We had to screening more than a thousand records classified as 'systematic reviews addressing other questions' in the HSE to obtain our required sample, as the majority of reviews in this category either adopted descriptive, narrative or qualitative approaches, or did not match our definition of an HSDR association review.

We ensured that we only include the latest systematic review whenever we identified more than one which covers overlapping topics. There may be some overlap in the studies included

within different systematic reviews, but we do not believe this would have significant impact on our findings as our study focuses on the overall features and methodology of the sampled systematic reviews rather than on individual studies included within them. We not only examined the proportion of systematic reviews which mentioned/assessed publication bias but also explored a number of factors which may influence these. Although the sample size of 200 reviews is still relatively small, as evident by the large confidence intervals for the ORs obtained from the multivariable logistic regression analyses, we were able to identify a few factors that may influence assessment of publication and outcome reporting bias in HSDR systematic reviews. We are aware that the variables which we examined may interact in various ways, as indicated by the changes in the estimated ORs between univariable and multivariable analyses for some of the variables examined. The relationships between the factors that could impact upon assessment of publication and outcome reporting bias in HSDR systematic reviews are intricate and will require further research to clarify.

The association between journal's endorsement and authors' use of reporting guidelines and assessment of publication bias may not have been characterised very accurately in our study. We classified journals based on endorsement of reporting guidelines as of 2018 but we were not able to determine if this has been the case as at the time the systematic review authors prepared/published their manuscripts. Notwithstanding, journal endorsement of such guidelines may be an indication of the journal's generic requirement of higher standard of reporting. Also, available reporting guidelines are mostly aimed at systematic reviews of intervention studies and authors of systematic reviews of association studies might not have considered that it was necessary to follow such guidelines, even if it was endorsed by the journal they published in. Alternatively, some authors might have used reporting guidelines during the preparation of their reviews without explicitly stating it.

HSE used AMSTAR to assess the quality of included systematic reviews. We also used the same tool to assess the quality of the five systematic reviews with missing AMSTAR scores in order to maintain consistency. However, AMSTAR was designed for quality assessment of systematic reviews of RCTs of interventions and therefore some of the items were not relevant for many of the systematic reviews in this study. An updated version of the tool, AMSTAR 2, was published in 2017 which includes items relevant to non-randomised studies and would have been more appropriate for assessing the qualities of the systematic reviews included in this study [33]. Another potential limitation of this study is that we only included systematic reviews of quantitative studies although HSDR involves a wide range of study design, including qualitative studies. However, we believe issues relating to publication bias and outcome reporting bias in qualitative research warrants separate investigation as the mechanisms and manifestation of such biases are likely to be different in qualitative research.

## Explanation of results and implications

Overall, the awareness of publication bias in quantitative HSDR reviews seems comparable to those reported for reviews in some other fields, although formal assessment of publication bias is less common especially in association reviews. Table 5 shows that the level of documenting awareness of publication bias by at least mentioning it was generally low in systematic reviews examined in previous studies in various fields of biomedical research, with a notable exception among systematic reviews of genetic association studies in which 70% mentioned publication bias. Unlike publication bias where many authors did discuss the potential implications even when they were not able to assess it, outcome reporting bias was only mentioned when it was assessed. However, mentioning of outcome reporting bias was lower than 30% across the

**Table 5. Findings from current and previous studies on assessment of publication and outcome reporting biases in systematic reviews of health literature.**

| Study and nature of systematic reviews examined | Searched grey literature/ unpublished studies* | Included meta-analysis | Mentioned publication bias | Formally assessed publication bias | Mentioned outcome reporting bias | Outcome reporting bias assessed |
|---|---|---|---|---|---|---|
| **Current review** | | | | | | |
| HSDR Intervention (n = 100) | 51% | 33% | 54% | 14% | 30% | 30% |
| HSDR association (n = 100) | 52% | 10% | 31% | 5% | 4% | 4% |
| **Li et al. 2015** [10] Health policy interventions (n = 99) | 67% judged to be comprehensive | 39% | 32%** | 9% | NR | NR |
| **Ziai et al. 2017** [30] High-impact clinical journals (n = 203) | 64% | NR | 61% | 33% | NR | NR |
| **Herrmann et al. 2017** [31] Clinical oncology (n = 182) | 27% conference abstract; 8% trial registries | NR | 40% | 28% | NR | NR |
| **Chapman et al. 2017** [32] High-impact surgical journals (n = 81 pre-PRISMA, n = 201 post-PRISMA) | Pre 71% Post 90% judged to be comprehensive | Pre 65% Post 78% | NR | Pre 39% Post 53% | NR | NR |
| **Page et al. 2016** [9] Biomedical literature (n = 300) | 16% conference abstract;19% trial registry | 63% | 47% | 31% | NR | 24% (n = 296) |
| **Song et al. 2010** [6] | | | | | | |
| Treatment effectiveness (n = 100) | 58% | 60% | 32% | 21% | 18% | NR |
| Diagnostic accuracy (n = 50) | 36% | 82% | 48% | 24% | 14% | NR |
| Epidemiological risk factors (n = 100) | 35% | 68% | 42% | 31% | 3% | NR |
| Genetic association (n = 50) | 10% | 96% | 70% | 54% | 16% | NR |
| **Kirkham et al. 2010** [34] Cochrane reviews of RCTs with well-defined primary outcome (n = 283) | NR | NR | NR | NR | 7% | NR |

*Figures are unlikely to be directly comparable as criteria used by different studies vary widely

**The actual figure is likely to be higher as this did not include situations in which "publication bias was not assessed for some reason".

NR: not reported.

board (17% in the current study), with very low rates observed in reviews of HSDR association studies (4% in the current study) and reviews of epidemiological risk factors (3% [6]).

A number of inter-related issues warrant further consideration when interpreting these findings and making recommendations. First, the research traditions and nature of evidence varies between different subject disciplines and may influence the perceived importance and relevance of considering publication and outcome reporting biases in the review process. These variations might have contributed to the apparently low prevalence of assessing and documenting these biases in HSDR reviews and wide variations observed in different disciplines. For example, we found that meta-analysis was conducted in only 33% of the HSDR intervention reviews. This is similar to 39% reported in a previous study of Cochrane reviews focusing on HSDR (health policy) interventions [10]. We found an even lower prevalence (10%) of including meta-analysis in HSDR association reviews. These figures are in contrast with at least 60% observed among both intervention and association reviews in clinical research (Table 5). There is a general recognition that HSDR requires consideration of multiple factors in complex health systems,[4] and that evidence generated from HSDR tends to be context-specific.[35–37] It is therefore possible that HSDR systematic reviews which evaluate intervention effects and associations, and particularly the latter which examine associations between the myriads of structure, process, outcome measures and contextual factors, may tend to adopt a more configurative, descriptive approach (as opposed to the more aggregative,

meta-analytical approach in reviews of various types of clinical research).[38] Since generating an overall estimate of a "true effect" is not the main focus, the issue of publication and outcome reporting biases may be perceived as unimportant or irrelevant in reviews adopting configurative approaches.

Furthermore, the diverse and context-specific nature of evidence in HSDR may have further impeded formal assessment of publication bias. Funnel plots and related techniques, the most commonly used methods, require that at least 10 studies of varied sample sizes that are addressing sufficiently similar questions and that have used compatible outcome measures to enable appropriate analyses [15]. In HSDR systematic reviews, the level of heterogeneity among included studies are often high and so reviewers are often not able to use these formal statistical techniques. Irrespective of the technical requirements, such statistical methods could only detect small study effects, which could be suggestive of publication bias but do not prove it, as several potential causes other than publication bias, such as issues related to study design, statistical artefact and chance, could also lead to the presence of small study effects [15].

With the inherent limitations of statistical tools, the most reliable way to directly assess publication and outcome reporting biases is by following up studies from protocol registration to see if the outcomes were subsequently published, as well as comparing the outcomes reported in protocols to those eventually reported in output publications. Mandatory registration of research protocols has been enforced among clinical studies on human subjects but not in other fields. The lack of prospective registration of study protocols has been a major barrier for evaluating publication and outcome reporting bias in HSDR as evidenced by the low prevalence of assessing these biases particularly among reviews of observational studies, e.g. 4% among HSDR association reviews in our study and 7% among epidemiological risk factor reviews examined by Song et al.[6]. Availability of pre-registered study protocols will potentially safeguard against publication and outcome reporting biases and also enable reviewers to assess those biases.

While pre-registration of study protocols is good research practice that should be encouraged irrespective of scientific disciplines, mandatory pre-registration of studies and their protocols in HSDR of different types of studies beyond clinical trials would require careful deliberation and assessment with regard to feasibility and practical value, weighing potential benefits against costs and potential harms. In the meantime, it is important to continue raising awareness of these biases and improving the levels of documenting the awareness when evidence from quantitative HSDR is synthesised. Our findings show that systematic reviews that report the use of a systematic review guideline are five times more likely than those that don't to include an assessment of publication bias. Another study which evaluated the impact of the PRISMA Statement on reporting in systematic reviews published in high-impact surgical journals reported that the proportion of systematic reviews which assessed publication bias was significantly higher after the publication of PRISMA (53%) compared to before PRISMA (39%) [32]. Methodological standards such as Cochrane Collaboration's Methodological Expectations of Cochrane Intervention Reviews (MECIR) and systematic reviews reporting guidelines such as PRISMA and MOOSE [18] are therefore likely to play an important role. Nevertheless, the sub-optimal level of documenting awareness found in this and other studies highlight that additional mechanisms may be required to enforce them. For example, although 70% of the systematic reviews in this study are published in journals which endorse systematic review guidelines, the use of such guidelines was only reported in 37% of the systematic reviews. Journal editors and peer reviewers can help ensure that review authors adhere to recommended guidelines which will in turn promote the consideration of publication bias.

All the reviews which assessed outcome reporting bas in the current study did so as part of quality assessment of individual studies, especially those that used the Cochrane risk of bias

tool [26]. Outcome reporting bias is a standard item in the current Cochrane risk of bias tool [26], which is most widely used in intervention reviews. However, this item is not included in tools commonly used for assessing observational studies such as the Newcastle-Ottawa scale [39]. This may contribute, in part, to the much higher proportion of intervention reviews that assessed outcome reporting bias compared to association reviews. Given that the risk of outcome reporting bias is substantially higher for observational studies, this is an important deficit which developers of quality assessment tools for observational studies need to address in the future.

Finally, the search and inclusion of grey/unpublished literature remain a potentially important strategy in minimising the potential effect of publication bias. In this study, 52% of the selected systematic reviews reported searching at least one source of grey literature. This is comparable to that which was reported (64%) in a recent audit of systematic reviews published in high ranking journals such as Journal of the American Medical Association, The British Medical Journal, Lancet, Annals of Internal Medicine and the Cochrane Database of Systematic Reviews [30]. The slightly higher value reported in the audit may be attributable to inclusion of only high impact journals. Our study further showed that reviewers who searched for grey literature do not necessarily assess/discuss the potential effect of publication bias. This suggests some review authors might have followed the good practice of searching the grey/ unpublished literature to ensure comprehensiveness without considering minimising publication bias as a rationale behind this. Alternatively, these authors may consider a comprehensive search as an ultimate strategy to mitigate potential publication bias and therefore deemed it unnecessary to assess and/or discuss the potential impact of publication bias. However, reviewers need to be aware that the search of grey literature alone is not enough to completely alleviate publication bias and it is often impractical to search all possible sources of grey literature. There is limited evidence suggesting that the quality and nature of data included in published HSDR studies differ from that included in grey literature [40]. Therefore, more empirical evidence is needed to guide future practice regarding search of grey/unpublished literature, taking into account the trade-off between biases averted and additional resources required.

## Conclusion

Publication and outcome reporting biases are not consistently considered/assessed in HSDR systematic reviews. Formal assessment of publication bias and outcome reporting biases may not always be possible until a comprehensive registration of HSDR studies and their protocols becomes available. Notwithstanding this, review authors could still consider and acknowledge the potential implications of these biases on the findings. Adherence to existing systematic review guidelines may improve the consistency in assessment of these biases. Including items for outcome reporting bias in future quality assessment tools of observational studies would also be beneficial. The findings of this study would enhance awareness of publication and outcome reporting biases in HSDR systematic reviews and inform future systematic review methodologies and reporting.

## Acknowledgments

We would like to thank the Health Systems Evidence for giving us access to the list of systematic reviews and Dr Kaelan Moat for facilitating it, and Alice Davis for her help in data checking. We also thank members of our Study Steering Committee for their helpful support and guidance through the project.

## Author Contributions

**Conceptualization:** Richard J. Lilford, Yen-Fu Chen.

**Data curation:** Abimbola A. Ayorinde.

**Formal analysis:** Abimbola A. Ayorinde.

**Funding acquisition:** Iestyn Williams, Russell Mannion, Fujian Song, Richard J. Lilford, Yen-Fu Chen.

**Investigation:** Abimbola A. Ayorinde, Iestyn Williams, Russell Mannion, Fujian Song, Magdalena Skrybant, Richard J. Lilford, Yen-Fu Chen.

**Methodology:** Abimbola A. Ayorinde, Iestyn Williams, Russell Mannion, Fujian Song, Magdalena Skrybant, Richard J. Lilford, Yen-Fu Chen.

**Project administration:** Abimbola A. Ayorinde, Yen-Fu Chen.

**Supervision:** Yen-Fu Chen.

**Validation:** Abimbola A. Ayorinde, Yen-Fu Chen.

**Writing – original draft:** Abimbola A. Ayorinde.

**Writing – review & editing:** Abimbola A. Ayorinde, Iestyn Williams, Russell Mannion, Fujian Song, Magdalena Skrybant, Richard J. Lilford, Yen-Fu Chen.

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
