## [Decision Letter · Decision Letter 0]

30 Oct 2019

PONE-D-19-25774

Assessment of publication bias and outcome reporting bias in systematic reviews of health services and delivery research: a meta-epidemiological study

PLOS ONE

Dear %Dr. Ayorinde,

Thank you for submitting your manuscript to PLOS ONE. After careful consideration, we feel that it has merit but does not fully meet PLOS ONE’s publication criteria as it currently stands. Therefore, we invite you to submit a revised version of the manuscript that addresses the points raised during the review process.

Reviewer 2 recommended adjusting the analysis for multiple comparisons. Your analysis is descriptive, so please ignore this comment. However, I think using statistical tests and dichotomizing p-values is not the best way for this descriptive analysis. I would suggest using univariate ORs with 95%-CIs similar to the other analyses. Another way maybe is not reporting any measure on statistical uncertainty in this analysis.

We would appreciate receiving your revised manuscript by 20. November 2019 To enhance the reproducibility of your results, we recommend that if applicable you deposit your laboratory protocols in protocols.io, where a protocol can be assigned its own identifier (DOI) such that it can be cited independently in the future. For instructions see: http://journals.plos.org/plosone/s/submission-guidelines#loc-laboratory-protocols

We look forward to receiving your revised manuscript.

Kind regards,

Tim Mathes

Academic Editor

PLOS ONE

Journal Requirements:

Reviewers' comments:

Reviewer's Responses to Questions

**Comments to the Author**

1. Is the manuscript technically sound, and do the data support the conclusions?

Reviewer #1: Yes

Reviewer #2: Yes

2. Has the statistical analysis been performed appropriately and rigorously? 

Reviewer #1: I Don't Know

Reviewer #2: No

3. Have the authors made all data underlying the findings in their manuscript fully available?

Reviewer #1: Yes

Reviewer #2: No

4. Is the manuscript presented in an intelligible fashion and written in standard English?

Reviewer #1: Yes

Reviewer #2: Yes

5. Review Comments to the Author

Reviewer #1: I wish to thank the editors for the opportunity to review this manuscript entitled “Assessment of publication bias and outcome reporting bias in systematic reviews of health services and delivery research: a meta-epidemiological study”. The authors have assessed how often publication bias and selective outcome reporting bias was mentioned and assessed in a random sample of systematic reviews of studies in the field of health services and delivery research. They have explained and reported their methods transparently.

I have the following comments for the authors:

1. Ll. 72-73: I suggest the authors explain in more detail why biased results of systematic reviews are a problem. E.g., as stated in their PROSPERO record: “This is important because HSDR frequently informs decisions at institutional and policy levels, and failure to recognise bias in evidence used to inform decisions could have substantial implications for population health and resource allocation.”

2. Ll. 97-116: I feel this part belongs to the method section and recommend to incorporate it in “eligibility criteria”.

3. Ll. 100-104 (“We examined key features”): In my view, this is repetitive and thus can be deleted.

4. Ll. 104-105: I am surprised that the study was registered in PROSPERO as it does not contain a health-related outcome. Nevertheless, I consider it very valuable that the methods have been pre-specified. I suggest that the authors delete the term “protocol” (because strictly speaking it is a registration record) and encourage them to update their PROSPERO record’s status to “completed but not published” (and remind to update it again to “completed and published” once their manuscript has been published). When doing so, they may want to explain important changes to their record since its initial version, i.e. the addition of two new authors, the changes in the search strategy already outlined in this manuscript, and the assessment with AMSTAR.

5. Ll. 137 and following (“Eligibility criteria”): I suggest to add the following information (taken from the PROSPERO record) at the end of this sub section: “Where more than one review within the initially selected samples cover overlapping interventions or associations, only the latest review will be retained to maintain the independence of observations (i.e. reduce overlap of included studies between reviews) and to capture the contemporary practice.” The authors should discuss the issue of potentially remaining overlap in the limitations section.

6. LL. 181-197 (data items): In their PROSPERO record, the authors stated that they would categorize the reviews according to the types of journal they were published in (medical journals, health services research and health policy journals, management and social science journals or others (including grey literature)). If that has been done, I suggest to add it as a data item and in the results. If not, the authors may want to describe that in the revision note for their updated PROSPERO record.

7. Ll. 213-214: If possible, the authors should explain in more detail the methodology that was used by HSE to assess the systematic reviews with AMSTAR (i.e. two people independently?). Furthermore, the authors should discuss in the limitations that the tool has several weaknesses (which eventually lead to the development of a new version of the tool https://www.bmj.com/content/bmj/358/bmj.j4008.full.pdf ).

8. Table 1: I was wondering if the authors have extracted the studies’ publication year. I consider this an important characteristic and strongly suggest that the authors add this in their analyses. For example, if it turned out that the association reviews were all published in 2007-2009 (before PRISMA was published), while most intervention reviews were published after 2009 (after PRISMA was published), this might also explain why fewer of them mentioned publication bias.

9. Table 1: “AMSTAR rating” should be specified, e.g. “Percentage of AMSTAR items rated positively” and “Journal endorses systematic review guideline” should be complemented by “(as of 2018)”

10. Ll. 273: Please quantify “often”, i.e. in how many review it was reported that the conditions for using funnel plots were not met? This also applies to ll. 323; please specify in how many reviews the lack of protocols was reported as a barrier to assessing outcome reporting bias. It may be worth to draw up a table that includes all reasons that systematic review authors named as barriers to assessing publication bias or outcome reporting bias.

11. Ll. 287 and 306: Commas are missing after the odds ratios.

12. Ll. 363 and following (Strengths and limitations): In addition to issues mentioned before, the authors should discuss that the item “endorsement of reporting guidelines” was assessed as of 2018, but that it is unclear if it has been the case at the time the systematic review authors have prepared their manuscripts. Furthermore, the authors should discuss that reporting guidance (at least by Cochrane and PRISMA) is aimed at systematic reviews of interventions. So, authors of systematic reviews on associations may not have followed it even if it was endorsed by the journal they have published in.

13. Ll. 367: Please provide citations when referring to “previous studies”.

I have the following discretional comments for the authors:

14. Ll. 60-62: I suggest to consistently use either no decimal or one decimal.

15. Ll. 188: I would reword “inclusion of meta-analyses” to “whether meta-analyses were performed” (and accordingly in other places in the manuscript) to stress that this is an active process.

16. Table 2: There is a space between 52 and % (Searched grey/unpublished literature).

17. Ll. 320: I suggest to reword “35% had at least one study …” to “35% identified at least one study”.

18. Ll. 470: I suggest to delete the word “interestingly”.

Reviewer #2: The paper is overall well-written, but requires a number of major and minor clarifications in the methods and definitions used, as well as adjusting for multiple comparisons in the analysis. A (very) short summary of the paper would be that it examines whether meta-analyses in the field of Health Services and Delivery Research (HSDR) report examining publiction and outcome reporting bias. This may be helpful to raise awareness of the need to examine these amongst meta-analysts in the HSDR field.

Major points:

- In the section 'Sampling Strategy' the authors report that the database healthsystemsevidence.org was used to pre-screen systematic reviews for inclusion, because the scope was too large for manual screening. But that the pre-screening is outsourced in this way does not obviate the need to describe how the corpus was defined. That is, what are the criteria for inclusion of systematic reviews in the healthsystemsevidence.org database? (search phrases etc)

- The authors run many hypothesis tests (Table 1: 15, Table 2: 9, Table 3: 10, Table 4: 10) and then interpret the _p_-values at face value. This overstates the evidence. I suggest correcting for multiple comparisons (e.g., using Bonferroni). Although it can sometimes be difficult to define what a "family" of tests are within which to correct the false-positive rate, in the authors case I think this is fairly straightforward as each table seems to present a different family of analyses (i.e., for Table 15 using a Bonferroni correction, alpha = nominal alpha/15). It is perhaps a little less clear what to do for Table 2:4 where the authors run both a set of univariable analyses and a multivariable analysis, here I would personally correct within each type of analysis. If the authors' are philisophically opposed to multiple comparison corrections for some reason, they need to at the very least interpret their results in view of the many tests they do, although the risk of overstating the evidence is much higher without formal corrections.

- Relatedly, the authors seem to largely avoid interpreting their own multivariable regressions? That is:

Line 360-361: "A higher proportion of the intervention review mentioned and assessed both biases compared to the association reviews". Indeed, if we only care about the margins, but if so what were all the regressions about? Main effects are not interesting when we have interactions. It is very strange to ignore the regressions when summarizing the results.

Line 374: "believe that collinearity is not a major issue judging from the broad consistency between the results of univariable and multivariable analyses". I disagree. For one, many many variables which were "significant" (assuming p <.05) in the univariable version become non-significant in the multivariable regressions. This is even more the case if an adjustment for multiple comparisons had been made. Broad consistency is a very questionable statement here. In addition, Table 1 present evidence *in favor* of collinearity between at least the variable "being an intervention review" and "including a meta-analysis". It is thus unclear whether the comparison between intervention and association reviews is relevant, or if difference are due to including (or not) a meta-analysis in the systematic review.

-Clarification of why the authors choose to compare "intervention" and "association" reviews, since this affects the presentation of most results. On lines 109-113, the authors state that:

[a] "We hypothesised that association studies may be more susceptible to publication and outcome reporting biases than intervention studies due to the exploratory nature of most association studies."

[b] "We therefore investigate whether the practice of assessing these biases and the findings of these assessments differ beween HSDR systematic reviews focusing on these two types of studies."

However, [b] does not follow from [a]. If the authors' hypothesis is that association studies are more susceptible to publication/outcome bias than intervention studies, then they should surely be researching this question? As it is, the authors presume their hypothesis to be true, and then essentially examine whether systematic reviewers are aware of this "truth". I could just as plausibly argue that, given the authors assumption that association reviews are more exploratory and intervention reviews more confirmatory, intervention studies should be more prone to publication bias and outcome switching because they are more invested in a given outcome than the exploratory association reviews. If so, it would make sense, again according to the authors' line of reasoning, that systematic reviews of intervention studies are more careful to assess potential biases.

Minor points/clarifications:

*The link to the Warwick Research Archive Portal is missing (currently written as XXXXX)

Line 80: I have not heard of MEDLINE previously and it is presented without an explanation. If this is something extremely well-known in the field of medical research this is fine, but otherwise please provide a short explanation of what it is.

Line 126: Very nice that the study was pre-registered, but I request a direct link to the pre-registration if possible. Few people check pre-registrations, so they need to be as accesible as possible.

Line 142-143: The categories used from healthsystemsevidence.org are "effectiveness" and "other". A discussion of how well these categories map onto those chosen by the authors(intervention and association) is lacking

Line 170-171: "such as those investigating effectiveness and cost effectiveness of clinical interventions" -> Please write out fully what was excluded rather that give examples

Line 176: "steering committee were consulted when necessary" -> under what circumstances was this necessary?

Line 187: there should be a reference here for the " minimum number of studies recommended.."

Line 190: The authors assume " that all Cochrane review adhered to [MECIR] standards even if not reported by authors". However, later the same assumption does not seem to be extended to journals with reporting standards (line 246), this needs to be discussed. In addition, that authors may have used guidelines without reporting their use is an important limitation which is not written out.

Line 209: Two coders did the coding, but no agreement level was reported. Please report Cohen's kappa or similar measure of inter-rater agreement.

AMSTAR: there is a new version of AMSTAR out. I understand the authors used the original measure for consistency with the healthsystemsevidence database, but considering there's an AMSTAR2 version, a critical discussion of the quality of AMSTAR is needed (perhaps in the limitations section). The AMSTAR webpage links several papers that could be a good starting point.

Line 227: How do you define "formal" assessment?

Results: A flowchart indicating the original sample and how many systematic review were excluded and for what reason to reach the final sample would improve clarity

Table 5: This table is misplaced. Don't present new information in the discussion section. Move to introduction or methods.Also explain what "NR" stands for.

Line 410-411: "40% of all reviews" in one study is compared with the current study's "10% for association review" -> The appropriate comparison is with % from all review in the current study. Comparing just association reviews is misleading.

Line 419: "the issue of publication and outcome biases may be perceived as unimportant or irrelevant in review adopting configurative approaches" -> It is suggested a large percentage of HSDR reviews are association reviews that use a configurative approach. This argument would be strengthened by adding what proportion of HSDR reviews are in fact association reviews (or "other", in the healthsystemsevidence database).

Line 420: Is there evidence HSDR is *more* diverse and context-specific than the fields that are compared against?

Line 425-426: Please write out some of these limitations rather than just referencing their existence. Also, to my knowledge these methods do *not* "indicate the presence or absence of publication bias", but of small-study effects.

6. PLOS authors have the option to publish the peer review history of their article (what does this mean?). If published, this will include your full peer review and any attached files.

Reviewer #1: Yes: Tanja Rombey

Reviewer #2: No

---

## [Author Response · Author response to Decision Letter 0]

18 Dec 2019

Editor's comment: Reviewer 2 recommended adjusting the analysis for multiple comparisons. Your analysis is descriptive, so please ignore this comment. However, I think using statistical tests and dichotomizing p-values is not the best way for this descriptive analysis. I would suggest using univariate ORs with 95%-CIs similar to the other analyses. Another way maybe is not reporting any measure on statistical uncertainty in this analysis.

Authors' reponse: We appreciate the editor for this suggestion. We have taken out p-values and presented % differences between groups and odds ratios with 95% Confidence intervals. 

Please see attached "Response to reviewers" document for responses to reviewers.

---

## [Editor Report · Decision Letter 1]

23 Dec 2019

Assessment of publication bias and outcome reporting bias in systematic reviews of health services and delivery research: a meta-epidemiological study

PONE-D-19-25774R1

Dear Dr. Ayorinde,

We are pleased to inform you that your manuscript has been judged scientifically suitable for publication and will be formally accepted for publication once it complies with all outstanding technical requirements.

With kind regards,

Tim Mathes

Academic Editor

PLOS ONE
---

## [Editor Report · Acceptance letter]

15 Jan 2020

PONE-D-19-25774R1 

Assessment of publication bias and outcome reporting bias in systematic reviews of health services and delivery research: a meta-epidemiological study 

Dear Dr. Ayorinde:

I am pleased to inform you that your manuscript has been deemed suitable for publication in PLOS ONE. Congratulations! Your manuscript is now with our production department. 

With kind regards,

on behalf of

Dr. Tim Mathes 

Academic Editor

PLOS ONE